# Physicochemical Characteristics and Bioactive Compounds of Different Types of Honey and Their Biological and Therapeutic Properties: A Comprehensive Review

**DOI:** 10.3390/antibiotics12020337

**Published:** 2023-02-06

**Authors:** Mohammad A. Al-Kafaween, Mohammad Alwahsh, Abu Bakar Mohd Hilmi, Dina H. Abulebdah

**Affiliations:** 1Department of Pharmacy, Faculty of Pharmacy, Al-Zaytoonah University of Jordan, Amman 11733, Jordan; 2Department of Biomedicine, Faculty of Health Sciences, Universiti Sultan Zainal Abidin, Kuala Nerus 21300, Malaysia

**Keywords:** natural honey, physicochemical properties, biological activities, bioactive compounds, polyphenols

## Abstract

Honey is considered to be a functional food with health-promoting properties. However, its potential health benefits can be affected by individual composition that varies between honey types. Although studies describing the health benefits of Tualang honey (TH), Kelulut honey (KH), and Sidr honey (SH) are scarce, these honey types showed a comparable therapeutic efficacy to Manuka honey (MH). The purpose of this review is to characterise the physicochemical, biological, and therapeutic properties of TH, KH, and SH. Findings showed that these honeys have antibacterial, antifungal, antiviral, antioxidant, antidiabetic, antiobesity, anticancer, anti-inflammatory and wound-healing properties and effects on the cardiovascular system, nervous system, and respiratory system. The physicochemical characteristics of TH, KH, and SH were compared with MH and discussed, and results showed that they have high-quality contents and excellent biological activity sources. Flavonoids and polyphenols, which act as antioxidants, are two main bioactive molecules present in honey. The activity of honey depends on the type of bee, sources of nectar, and the geographic region where the bees are established. In conclusion, TH, KH, and SH could be considered as natural therapeutic agents for various medicinal purposes compared with MH. Therefore, TH, KH, and SH have a great potential to be developed for modern medicinal use.

## 1. Introduction

Honey is a natural sweetener with a complex chemical composition and health/promoting properties [1,2]. Bees collect nectar from plants and/or insect excretions and produce honey, which has been revered for centuries for its nutritional and therapeutic properties [3]. Honey has been resurrected as a therapy for burns, gastrointestinal diseases, asthma, infected wounds, and skin ulcers in humans, and in animal medicine [4,5]. Honey contains several constituents of small amounts, such as minerals, free amino acids, proteins, vitamins, enzymes, organic acids, flavonoids, phenolic acids, and other organic acids in addition to other phytochemicals compounds [6,7]. The amount of these components is determined by several factors, including the honey’s geographical origin, floral source, meteorological circumstances, any treatments applied [8], and seasonality [9]. Honey’s composition can be affected by processing, handling, and storage [10]. The quality of honey also depends on floral resources and the treatment of the beekeepers [11]. Honey’s botanical and geographical origins have traditionally been determined by evaluating pollen quality and quantity and organoleptic and physicochemical testing. In addition, data derived from the sensory profile, bioactive components, and novel methods of investigation should be added to this information [7,12,13]. Water content, sugar reduction, sucrose, insoluble matter, ash, free acid, pH, electrical conductivity, specific rotation, and sensory and microbiological properties are the basis for the quality assessment of honey [14,15]. Honey’s components have a variety of beneficial biological actions, such as antibacterial, antiviral, antifungal, antioxidant, antidiabetic, antitumor, anti-inflammatory, and anticancer activity honey [16,17,18] (Figure 1). Various studies have demonstrated that antioxidant activity highly correlates to total phenolic levels [19]. Moreover, darker honey has been reported to have a higher total phenolic content and thus more significant antioxidant activity [20]. Honey’s composition includes various components, including hydrogen peroxide and polyphenols, and is also strongly linked to antibacterial activity [21,22]. The ability of honey to fight different types of microorganisms is determined by various variables, including the kind and natural structure of the nectar and the environmental circumstances in which the bees were raised [23,24]. The phenolic and flavonoid chemicals that make up honey are thought to be responsible for most of its biological activity. According to previous research, the activity of honey is influenced by the bioavailability of different phytochemical components as well as how they are absorbed and metabolized [18,25]. The flavonoids are largely water-soluble natural chemical compounds with low molecular weight. When flavonoids are not linked to sugars, they are referred to as aglycones [26,27,28,29,30,31,32,33]. In general, these chemicals include at least two phenolic groups (OH) and are frequently coupled with sugars (glycosides) [34,35,36,37,38,39,40,41,42,43,44,45]. The term “phenolic acids” refers to compounds that have a phenolic ring and at least one organic carboxylic acid function. They can be classified into three groups based on their structural types: C_6_–C_3_ compounds (such as p-coumaric, ferulic, and caffeic acids), C_6_–C_2_ compounds (such as acetophenones and phenylacetic acids), and C_6_–C_1_ compounds (e.g., syringic, vanillic and gallic acid). Most of these substances are typically linked to the cellulose and lignin that serve as the plant’s structural foundation, as well as to other classes of organic molecules such as glucose, other sugars, and flavonoids [46]. Additionally, some phenolic chemicals found in honey, including acaetin, caffeic acid, quercetin, galangin, and kaempferol, may hold potential as medicines for the treatment of cardiovascular disorders [47]. According to numerous research, flavonoids, which are crucial in reducing oxidative stress, are also necessary for honey’s antioxidant potential [13,48,49,50,51,52,53,54,55,56,57,58,59,60]. An updated study to comprehensively analyse the benefits of TH, KH, and SH is still required compared with other types of honey such as MH which has been extensively reviewed. With the recent growth in the body of literature for these kinds of honey, assessment of MH, TH, KH, and SH is needed to avoid future cross-studies or unnecessary research due to missed reviews of the existing research. In this review, relevant studies related to the medicinal properties, health benefits and physicochemical properties of MH, TH, KH, and SH were identified and critically analysed. The findings showed that most studies reported the health advantages of honey consumption, whereas some reported disadvantages or no significant changes upon consumption. Furthermore, analyses of physiochemical-related research revealed that MH, TH, KH, and SH have high-quality contents and are excellent sources of antioxidants.

## 2. Methods of Review

A literature search was conducted to identify previous articles illustrating the physiochemical and phytochemical of Manuka honey, Tualang honey, Kelulut honey, and Sidr honey and efficacy in the cure of diseases. Several databases were queried, including Web of Science, Scopus, Science Direct and PubMed. A literature search was performed by combining the following set of keywords: Manuka honey, Tualang honey, Kelulut honey, and Sidr honey. In addition, a literature search was undertaken to identify and map out relevant and pertinent articles related to the physicochemical, biological and therapeutic properties and health benefits of Manuka honey, Tualang honey, Kelulut honey, and Sidr honey. The present review covers a 52-year period which includes publications from 1970 to 2022. Initial searches yielded nearly 320 results. The abstracts of these publications were reviewed to confirm applicability. After further exclusion criteria (non-English language and manuscripts not available in full text), 243 articles remained. 

## 3. Physicochemical Properties and Composition of MH, TH, KH and SH

Honey has a wide range of physicochemical properties depending on its botanical and geographical origins and compositions that subsequently affect its biological capabilities [29,62,63,64,65,66,67,68,69,70,71,72,73,74,75,76,77,78,79,80,81,82,83,84,85,86,87,88,89,90,91,92,93,94,95,96,97,98,99,100,101,102,103,104,105,106,107,108,109,110,111,112,113,114,115,116,117,118,119,120,121,122,123,124,125,126,127,128,129,130,131,132,133,134,135,136,137,138,139,140,141,142,143,144,145,146,147]. Honey is defined as the natural sweet material generated by the Apies mellifera bees from the nectar of plants in European Union Council Directive 2001/110/EC [30]. Table 1 shows the physicochemical characterisation of MH TH, KH and SH extracted from various studies. Several studies have been conducted to look at the physical and chemical properties of MH TH, KH, and SH. On the other hand, MH TH, KH, and SH mostly complied with the accepted range by the two most common legislation of honey criteria and standards referred to as the European Honey Legislation and Codex Alimentarius Standards for Honey [29,31,32]. The primary quality indicators for honey include moisture content, sucrose content, reducing sugars content, pH value, electrical conductivity, ash content, free acidity, diastase activity, and hydroxymethylfurfural (HMF) content [36,37]. In terms of honey colour characteristics, MH is categorized as light-colored honey, TH is dark brown honey, KH is amber brown, and SH is dark-colored honey [29,38,43,148]. In this review most of the studies reported that TH and KH contained more than 20% moisture content, thus violating European Honey Legislation and Codex Alimentarius Standards. Nonetheless, honey samples from tropical countries, such as Malaysia, typically have higher moisture content, which could be due to the rainy season all over the year. Therefore, Malaysia’s honey is always first treated by evaporation to reduce the water content, thereby simultaneously increasing the honey quality [11,29,31,35,41,51,55,56]. MH contained 18.7% moisture content [41,43,50] and SH contained 13.5–20.67% moisture content [11,31,55,56]. The low pH of MH is approximately similar to TH and less than to KH and SH. TH is more acidic than KH and SH [41]. Darker honey typically has a higher conductivity, whereas brighter honey typically has a lower conductivity [33,34]. The pH values of TH, KH, and SH were reported to be in the range 3.14–4, 2.76–4.66 and 3.90–5.2, respectively, compared with MH (3.20–4.21) [11,29,31,35,41,42,43,51,55,56,57,59]. The electrical conductivity of TH, KH, and SH reported in this review is in a broad range of 0.75–1.37 mS/cm, 0.26–8.77 mS/cm, and 0.53 mS/cm, respectively, compared with 0.53 mS/cm to MH. Additionally, the four honey types met the sugar content requirements set forth by the Codex Alimentarius Standards and the European Honey Legislation. According to the European Honey Legislation and the Codex Alimentarius Standards, honey moisture should be less than 20%, with glucose and fructose composition of more than 60 g/100 g, sucrose content of not more than 5 g/100 g and electrical conductivity of not more than 0.8 mS/cm. According to the Malaysian Standard Kelulut, the raw honey moisture content must be less than 35 g/100 g, with a pH of less than 3.8 and 5-hydroxymethylfurfural of less than 30 mg/kg [29]. In addition, MH reported to has high protein content (g/kg) in the range 5.02–5.06 (g/kg) compared to 3.6–6.6 (g/kg) TH, 3.9–8.5 (g/kg) KH, and 1.5–4.09 (g/kg) SH [11,29,31,35,41,43,50,51,52,53,54,55,56]. Most bacteria grow in a neutral and mildly alkaline environment, whereas yeasts and moulds could grow in an acidic environment (pH = 4.0–4.5). Conversely, the pH values of honey are neither those needed for bacteria nor yeast growth [38]. This is of great importance during storage, as they influence the texture, stability, and shelf-life of honey [11,29,35,39,43,50,51,57,59,60,62]. The low protein content and high carbon-to-nitrogen ratio of honey are not conducive to microbial growth, nor is the acidity of honey. The low redox potential of honey (which is due to its high content of reducing sugars) discourages growth of molds and aerobic bacteria, whereas the viscosity of honey opposes convection currents and limits the entry of dissolved oxygen. As the osmotic pressure is high, the microbes shrivel as water flows out of their cells into the surrounding honey [29,39]. Various factors, such as storage, time, temperature, water content and concentration of ions and minerals, were reported to contribute to the electrical conductivity of honey [11,35]. A comparison of the physicochemical characteristics of TH, KH, and SH with that of MH is presented in Table 1.

Honey is mostly composed of fructose (35.6–41.8 g), glucose (25.4–28.1 g), sucrose (0.23–1.21 g) and Maltose (1.8–2.7 g) and other sugars [29,149,150,151,152,153,154,155,156,157,158,159,160,161,162,163,164,165,166,167,168,169,170,171,172,173,174,175,176,177,178,179]. It includes around 180 different compounds, including amino acids, vitamins, minerals, and enzymes. The composition varies according to the floral source and origin [29,179]. The concentration of sucrose (g/100 g) in MH and SH was higher than that in TH and KH [29,55,64,65], and the concentration of glucose (g/100 g) in MH and TH was higher than that in KH and SH [29,55,61,65]. In addition, the concentration of fructose (g/100 g) in MH and TH was higher than that in KH and SH [29,55,61,65], and the concentration of maltose (g/100 g) in TH and KH was higher than that in MH and SH [29,61,65]. Moreover, the protein content of MH (5.02–5.06 (g/kg) was higher than the values reported for TH (3.6–6.6 (g/kg), KH (3.9–8.5 (g/kg), and SH (1.5–4.09 (g/kg) [29,55,64,65]. Among the major and minor elements found in honey, potassium (K) is found in the highest concentrations, followed by sodium (Na), calcium (Ca), and magnesium (Mg). Furthermore, because sugars are its primary constituents, honey’s physical characteristics and behavior are attributed to sugars. Sugar tests will indicate its sweetness due to its high sugar content, with fructose being the most abundant sugar. The concentration of sodium in MH, KH and SH was lower than the values reported for TH and the concentration of potassium and calcium in SH was higher than the values reported for MH, TH, and KH [29,55,61,66]. Additionally, the concentration of magnesium in SH was higher than the values reported for MH, TH, and KH [29,55,61,66]. Minor constituents such as flavor compounds, minerals, acids, pigments, and phenols play a significant role in distinguishing each variety of honey [30,36,40,41]. Honey is a natural source of flavonoids, phenolic acids, and phenolic acid derivatives [44]. MH has higher total phenolic content (429.61 (mg/kg)) [64] than TH (251.7–1103.94 (mg/kg)) [29], KH (477.30–614.7 (mg/kg)) [29], and SH (212.4–520.34 (mg/kg)) [65]. In addition, the total flavonoid content in MH (97.62) [64] was higher than in TH (49.04–185) [29], KH (36.3) [29], and SH (42.5) [67]. A total of seven phenolic acids (caffeic, gallic, syringic, vanillic, p-coumaric, benzoic, and trans-cinnamic acids) and six flavonoids (apigenin, kaempferol, luteolin, naringenin, naringin, and catechin) are found in TH. Additionally, a total of four phenolic acids (syringic, gallic, ferulic and caffeic acids) and eight flavonoids (chrysin, galangin, isorhamnetin, kaempferol, luteolin, pinobanksin, pinocembrin, and quercetin) are found in MH. A total of nine phenolic acids (gallic, syringic, vanillic, 3 4-dihydroxybenzoic, 4-hydroxybenzoic, p-coumaric, cinnamic, salicylic, cis-trans-Abscisic acids) and three flavonoids (luteolin, naringenin, and taxifolin) are found in KH. A total of six phenolic acids (gallic, salicylic, chlorogenic and tannic acids) and five flavonoids (catechin and quercetin) are found in SH [28,29,41]. TH contains more phenolic acids and flavonoids than MH, KH and SH [41]. Some compounds found in TH previously not reported in other honeys include stearic acids, 2-cyclopentene-1,4,-dione, 2[3H]-furanone or dihydro-butyrolactone, gamma-crotonolactone or 2[5H]-furanone, 2-hydroxy-2-cyclopenten1-one, hyacinthin, 2,4-dihydroxy-2,5-dimethyl3[2H]-furan-3-one, and phenylethanol [41,179]. The details of the various compounds present in MH, TH, KH, and SH are summarised in Table 2 and Figure 2. 

Generally, honey is rich in phenolic compounds, which act as natural antioxidants and are becoming increasingly popular because of their potential role in contributing to human health [29,43,47,68,179]. In this review, Table 3 shows some of phenolic compounds with their different potential health benefits found in honey. A wide range of phenolic constituents are present in honey, including quercetin, caffeic acid, gallic acid, catechin, apigenin and kaempferol, which have promising effect in the treatment of cardiovascular diseases [29,43,47,68,179]. Many epidemiological studies have shown that regular intake of phenolic compounds is associated with reduced risk of heart diseases. In coronary heart disease, the protective effects of phenolic compounds include being antithrombotic, anti-ischemic, anti-oxidant, and vasorelaxant [29,43,47,68,179]. It is suggested that flavonoids decrease the risk of coronary heart disease by three major actions: improving coronary vasodilatation, decreasing the ability of platelets in the blood to clot, and preventing low-density lipoproteins (LDLs) from oxidizing [29,43,47,68,179]. Cell viability of fibroblast-like synoviocytes (FLS) from patients with rheumatoid arthritis (RA) was significantly decreased by treatment with 10 or more μM of gallic acid. Treatment with 0.1 and 1 μM of gallic acid also showed in a significant increase in caspase-3 activity and regulated the production of Bcl-2, Bax, p53, and pAkt. The mRNA expression levels of pro-inflammatory cytokines (IL-1β, IL-6), chemokines (CCL-2/MCP-1, CCL-7/MCP-3), cyclooxygenase-2, and matrix metalloproteinase-9 from RA FLS were suppressed by the gallic acid treatment in a dose-dependent manner [29,43,47,68,179]. The phenolic compounds in honey such as p-hydroxibenzoic acid, cinnamic acid, naringenin, pinocembrin, and chrysin showed antimicrobial activity [29,43,47,68,179]. Additionally, caffeic acid exhibits a significant potential as an antidiabetic agent by suppressing a progression of type 2 diabetic states that is suggested by an attenuation of hepatic glucose output and enhancement of adipocyte glucose uptake, insulin secretion, and antioxidant capacity [29,43,47,68,179]. In addition, protocatechuic and p-hydroxybenzoic acid exhibit significant antioxidant, anticancer and antiatherogenic activities [29,43,47,68,179]. The chrysin decreased lipid peroxide, reduced the increased activities of superoxide dismutase, and attenuated the decreased activities of glutathione peroxidase in 2VO rats and Quercetin-3-O-rhamnoside showed moderate antitumor activity [29,43,47,68,179]. 

## 4. Therapeutic Proprieties of MH, TH, KH and SH

The therapeutic benefits of MH, TH, KH, and SH are discussed in this section based on different properties. Additionally, literature on the therapeutic benefits of MH, TH, KH, and SH were obtained. These articles covered in vitro, in vivo, and ex vivo studies, and human clinical trials.

### 4.1. Oxidative Stress, Antioxidant and Anti-Inflammatory Properties

Oxidative stress (OS) is characterized as an imbalance in favor of oxidants over antioxidants. The OS causes oxidative damage, which can impair a variety of physiological activities. OS is the basis of structural and functional damage to the main biomolecules such as nucleic acids, lipids, and proteins. In fact, these injuries lead to the development of many diseases, such as cancer, metabolic disorders, and cardiovascular dysfunctions. The imbalance created between the production of free radicals and antioxidant defense can occur not only in pathological situations, but also in some physiological conditions such as intense physical activity [179]. The primary oxidants in biological systems are free radicals and reactive oxygen species (ROS). They are also implicated in aging and the beginning of many diseases [71]. ROS and free radicals are physiologically formed in several cellular biochemical events that occur in the body, such as in mitochondria for aerobic oxygen synthesis, fatty acid metabolism, medication metabolization, and immune system activity [72,73,74]. Free radicals, on the other hand, can be created by external sources such as pollution, poor lifestyle choices, UV rays, ionizing radiation, and psychophysical stress from strenuous physical exercise [75]. Antioxidants are compounds that may transfer an electron to free radicals, thus neutralizing, reducing, or eliminating their capacity to harm cells and key macromolecules including nucleic acids, proteins, and lipids [76]. Oxidative stress is more likely to cause chronic or degenerative disorders. The antioxidant properties of honey may help in reducing oxidative damage and improving brain cell structure and integrity [77,78]. Honey generally contains various kinds of phytochemical with high phenolic and flavonoid content, thus contributing to its high antioxidant activity [28,29]. Many studies have indicated that honey’s antioxidant ability is significantly related to the concentration of its phenolic component [64,68,79]. The present review also found that TH, KH, and SH have a high content of phenolics and flavonoids compared with MH. Amongst the reviewed studies comparing the antioxidant properties of TH and KH, one study showed that KH has the highest antioxidant properties in ranking after MH [57]. A previous study found that TH has higher antioxidant properties than MH and KH [29]. In addition, MH appears to play a protective function against oxidative damage in an in vivo model, lowering DNA damage, the level of malondialdehyde, and glutathione peroxidase activity in the livers of both young and middle-aged groups of rats. However, the glutathione peroxidase activity was increased in the erythrocytes of middle-aged rats given MH supplementation. The catalase activity was reduced in the liver and erythrocytes of both young and middle-aged rats given supplementation [80]. A previous study reported that the pancreases of diabetic control rats showed significantly increased levels of malondialdehyde (MDA) and up-regulation of superoxide dismutase (SOD) and glutathione peroxidase (GPx) activities and catalase (CAT) activity was significantly reduced, whereas glutathione-S-transferase (GST) and glutathione reductase (GR) activities remained unchanged in the pancreases of diabetic rats after being treated with TH (honey-treated diabetic rats had significantly (*p* < 0.05) reduced blood glucose levels (8.8 (5.8) mmol/L; median (interquartile range)) compared with the diabetic control rats (17.9 (2.6) mmol/L]) [62]. The DPPH radical scavenging activity was expressed in EC50 (mg/mL), where the minimum amount needed to scavenge 50% of the DPPH free radical [51,87]. A previous study showed that MH has higher antioxidant potential compared with TH, KH, and other types of honey (MH showed the lowest EC_50_ value (116.05 mg/mL), followed by TH (341.25 mg/mL) and KH (329.89 mg/mL) [51]. A study showed that TH increases the effectiveness of glibenclamide and metformin in preventing oxidative stress and damage in the pancreases of diabetic rats and reduces oxidative stress in diabetic rats’ kidneys [81,82]. In addition, TH enhanced the migration of human corneal epithelial progenitor cells and cellular resistance to H_2_O_2_-induced oxidative stress, according to an in vitro study [83]. In contrast, there was no discernible difference between the TH treated group and the group that received conventional treatment in terms of inflammatory feature or antioxidant status, according to a study looking at the anti-inflammatory and antioxidant effects of TH in alkali injury on rabbit eyes [84]. A previous study reported that KH reduced oxidative stress by reducing lipid peroxidation and increasing SOD, and also maintained bone structure, increased the number of osteoblasts, and reduced the number of osteoclasts. Therefore, the study suggested that KH could be used as a prophylactic agent to prevent glucocorticoid-induced osteoporosis [85]. Additionally, KH supplementation protected sperm and testicular oxidative damage in streptozotocin-induced diabetic rats [86]. A study by Saeed et al. (2021) demonstrated that SH exhibited considerable variations with reference to their level of total phenolic content (TPC) (98.2–432.91 mg/kg), total flavonoid content (TFC) (49.9–202.0 mg/kg), radical-scavenging activity (7.6–72.6 mg/mL), and ascorbic acid (28.1–161.2 mg/kg), as well as total carotenoid content (TCC) (13.2–36.2 mg/kg). A strong significant correlation between biochemical parameters and radical scavenging activity was found, and the study indicated that SH can be considered a good source of antioxidant and biochemical compounds [87]. A previous study showed that TH decreased neuroinflammation by lowering the elevation of TNF-, IL-1, glial fibrillary acidic protein, allograft inflammatory factor 1, and COX-2 in the rat cerebral cortex, cerebellum, and brainstem in kainic acid (KA)-induced status epilepticus rats [103]. Another study also reported a reduction in TNF-α, IL6, and IFN-γ in the brain homogenates of a TH-treated chronic stress rat model [88]. However, the anti-inflammatory properties of TH could not be translated to humans as a randomised controlled study demonstrated that TH supplementation has opposite effects on TNFα and highly sensitive C-reactive protein, indicating the inconclusive effect of honey on inflammation amongst chronic smokers; thus, further studies are needed on other inflammatory markers or other study populations [89]. One study showed that MH provided protection against trinitro-benzo-sulphonic acid induced colonic damage, reduced colonic inflammation also restored lipid peroxidation and improvement of antioxidant parameters [90]. Minden et al. (2020) reported that 0.5% of MH solution significantly increased the release of CXCL8/IL-8, CCL2/MCP-1, CCL4/MIP-1β, CCL20/MIP-3α, IL-4, IL-1ra, and FGF-13 while reducing Proteinase 3 release in the anti-inflammatory-stimulated models and 3% of MH solution significantly increased the release of TNF-α and CXCL8/IL-8 while reducing the release of all other analytes. These findings demonstrated the variable effects of MH on the release of cytokines, chemokines, and matrix-degrading enzymes of this model of neutrophil anti-inflammatory activity [91]. Another study showed that pre-treatment with MH markedly inhibited LPS induced ROS and nitrite accumulation and increased the protection against cellular biomolecules such as lipids, proteins, and DNA and stimulation by LPS suppressed both antioxidant enzyme activities and expressions, and Keap1-Nrf2 signaling pathway which was significantly increased in the presence of MH also after MH treatment the expression of pro-inflammatory cytokines, such as TNF-α, IL-1β, and IL-6, and other inflammatory mediators (iNOS) were suppressed [92]. Moreover, MH also inhibited the expression of TLR4/NF-кB via IкB phosphorylation in LPS-stressed RAW 264.7 macrophages [92]. KH has been proven to possess various pharmacological properties such as antioxidant and anti-inflammatory [93,94]. Since KH has strong antioxidant activities, it could be one of the potential chemopreventive agents from natural resources [93]. According to multiple studies, the quantity of phenolic and flavonoid groups affects the antioxidant activity of various types of honey produced from various nations [32,95,96]. MH acted as a natural agent for preventing oxidative and inflammatory-related diseases more than TH and SH.

### 4.2. Antibacterial, Antiviral and Antifungal Properties

Honey has been used as an antibacterial agent since ancient times. Table 4 describes the microorganisms that are sensitive to MH, TH KH and SH. It seems to act on both Gram-positive (G+) and Gram-negative (G−) bacteria, although the first are more sensitive. All the studies, summarized in Table 4, used the agar disk-diffusion and Broth dilution tests to determine the minimum inhibitory concentration (MIC) and minimum bactericidal concentration (MBC) of various types of honey with diverse bacterial agents. Numerous studies and research have been conducted in laboratories on the biological properties of honey such as antibacterial, antifungal, antiviral, and antiprotozoal [97,98,99]. The biological activity of honey is one of the most important features that distinguish it from other natural products, which makes it important in the medical and therapeutic fields [100,101,102]. Nowadays, honey is effective against more than 60 bacterial species, and its antibacterial activity is dependent on the type of honey. Numerous studies have reported that honey has antibacterial and antibiofilm activity against a wide range of both Gram-positive and Gram-negative bacteria, including; *Streptococcus pyogenes*, *Mycobacterium*, *Escherichia coli*, *Pseudomonas aeruginosa*, *Salmonella typhi*, *Salmonella paratyphi*, *Salmonella enterocolitis*, *Shigella dysenteriae*, *Pseudomonas aeruginosa*, *Mycobacterium tuberculosis*, methicillin-resistant *Staphylococcus aureus*, *Streptococcus pneumonia*, *Streptococcus agalactiae*, and *Shigella flexneri*. Among the activity of all types of honey, the inhibiting effect of MH, at low concentrations (2–6%), on the planktonic and biofilm of both Gram-positive and Gram-negative bacteria, was greater than other types of honey, including TH, KH, and SH; moreover, the large number of studies of MH has shown that it has broad spectrum activity against pathogenic bacteria [58,103,104,105,106,107,108,109,110]. TH was the most comparable to MH in terms of antibacterial potency, followed by KH and SH [24,31,104,107,111,112,113,114]. A previous study revealed that when MH and TH dressings were tested against Gram-negative bacteria on the burn wound, the results of two comparative tests on antibacterial potency were equivalent, but TH was less effective than standard care products as silver-based dressings or medical grade and MH dressings against Gram-positive bacteria [115]. TH, KH, and SH exhibited variable activities against different microorganisms, but they were within the same range as those for MH, suggesting that TH could potentially be used as an alternative therapeutic agent against certain microorganisms, particularly *Stenotrophomonas maltophilia* and *Acinetobacter baumannii* [31,107,116]. Honey has also been found in some studies to have antiviral action [29]. Küçük et al. (2007) and Mohd et al. (2021) demonstrated that honey has been known to reduce viral load since the 19th century [29,117]. It was evaluated that the in vitro effect of MH and Clover honey in human malignant melanoma cells (MeWo)-infected with varicella Zoster virus (VZV) isolated from a Zoster vesicle. The results showed a reduction of the viral plaques after the treatment of the cells with both types of honey [118]. A similar effect has also been proven in Madin-Darby canine kidney (MDCK) cells infected with influenza virus (H1N1), treated with different types of honey (MH, Renge honey, and Acacia honey). The plaque inhibition assay has been carried out showing a higher antiviral activity of Manuka honey, compared with the other types of honey, and the synergistic effects with some antiviral drugs [119,120]. Another study reported that among the five honey types (Manuka honey (MH), clover honey, acacia honey, rosemary honey, and milk vetch honey), the anti-HIV-1 RT activity of MH was the strongest and was associated with its constituents, 2-methoxybenzoic acid (2-MBA) and methylglyoxal (MGO) and MH inhibited the activity of HIV-1 RT in a dose-dependent manner and the IC_50_ value was approximately 14.8 mg/mL and the study revealed that the inhibitory effect of MH on the HIV-1 RT activity is mediated by multiple constituents with different physical and chemical properties as mentioned above [121]. It has also been demonstrated an antifungal activity of honey towards different kinds of Candida infections (*Candida albicans*, *Candida glabrata*, *Candida dubliniensis*, *Candida tropicalis*, *Candida krusei*, and *Candida parapsilosis*), and on *Rhodotorula* sp., evaluating the MIC and using the agar disk-diffusion test [122,123]. Study by Guttentag et al. (2021) indicated that jarrah honey has unique antifungal attributes that work to inhibit and kill dermatophytes, making it a potentially promising candidate for the treatment of tineas [124]. This variation was linked to the botanic origin of the honey, with multi-floral honey having greater phenolic concentrations than monofloral honey [125]. The antibacterial, antiviral, antifungal, antioxidant, and anti-inflammatory activities of honey are noteworthy due to phenolic compounds, especially flavonoids, with the quality of polyphenols being more important than their quantity [18,28,32,63,126]. In addition, honey’s potent antibacterial properties are related to its high osmolarity, acidity, H_2_O_2_ content, and non-peroxide component composition [127]. H_2_O_2_ is produced when glucose oxidase hydrolyzes glucose, which can only happen when honey is diluted. Relative glucose oxidase levels produced by bees and catalase derived from flowers are used to calculate H_2_O_2_ levels [128]. The antibacterial activities of honey at this stage are due to strong osmotic pressures and high acidity in pure honey [127]. MH possess high non-peroxide antibacterial activity that could retain the antibacterial activity even after treatment with catalase [29,129]. They are known as active non-peroxide honey, containing various non-peroxide components that possess antibacterial actions [130,131]. These components include phenolic acids, flavonoids, methylglyoxal, and methyl syringate [29,127]. Finally, honey’s strong antibacterial ability is connected to an improvement in gut microbial balance due to its high number of oligosaccharides, which function as a substrate for the growth of prebiotic microorganisms. One research found that the addition of different types of honey increased the vitality and development rate of Lactobacilli and Bifidobacteria in the gut microbiota balance [132].

### 4.3. Anticancer Properties

The potential effects on cancer have been studied in terms of prevention, progression, and therapy. The majority of the investigations are in vitro and have been conducted on various cell lines and types of honey. Some studies have also been carried out in vivo on mice/rats, inducing or transplanting the tumor [157]. Honey acts at different stages of cancer, on the initiation, proliferation, and progression. Its antitumoral effects are generally attributed to different mechanisms, such as the induction of apoptosis, cell cycle arrest, the modulation of oxidative stress, the amelioration of inflammation, the induction of mitochondrial outer membrane permeabilization (MOMP), and the inhibition of angiogenesis (Figure 3) [18,29]. The majority of the research on MH, TH, KH, and SH concentrated on their anticancer effects in a variety of models and pathways. Investigations involved a variety of subjects, including cell culture, animal, and human studies. A summary of the anticancer properties for MH, TH, KH, and SH is presented in Table 5, whereas their anticancer pathway is illustrated in Figure 3. The impact of MH and TH on breast cancer has also been examined in an in vivo investigation. It was shown that there was a decrease in tumor grade, estrogenic activity, and hematological parameters. Additionally, it has been shown that the expression of certain proteins involved in the inflammatory pathway, such as TNF- and COX-2, as well as pro-apoptotic proteins such as Caspase 9 and p53, has increased [158]. In another study, the activity of MH against HCT-116, human colon cancer cells, and the LoVo metastatic cell line was examined. On both cell lines, they observed an increase in intracellular ROS generation as well as an inhibitory effect on cell proliferation. These results show that the cytotoxic impact of MH may be connected to the quantity of polyphenols present in this matrix as the cytotoxic effect of STH was emphasized as being more pronounced, which exhibited a higher number of phytochemicals and antioxidant capabilities [18,159]. In one vivo study, MH induced a strong proapoptotic activity in a dose- and time-dependent way after being intravenously delivered to mice with murine melanoma tumor cells (B16F1), decreasing the final tumor volume. Additionally, mice which received MH in addition to the chemotherapy drug (Taxol) lived longer than mice that simply received the chemotherapeutic agent [160]. Another study demonstrated that MH has anticancer activity against breast cancer MCF-7 cells [161]. Three in-vivo studies using a breast cancer model showed that TH had anticancer potential [158,162,163]. These studies have shown that TH’s anticancer efficacy was demonstrated by modulating five aspects: tumor development, tumor grading, and haematologic, oestrogenic, and apoptotic activities. In a rat breast cancer model, TH-treated rats had a lower growth rate, tumor size, weight, and tumor multiplicity than untreated controls. [158,162,163]. Histologically, breast cancer rats treated with TH were mostly graded I and II compared with the untreated control [158,162]. Moreover, TH increased proapoptotic protein expression (Apaf-1, caspase-9, IFN-γ, IFNGR1, and p53), whereas it decreased antiapoptotic protein expression (ESR1, TNF-α, COX-2, and Bcl-xL) [158,162,163]. Ahmed et al. (2017) reported that TH treatment was effective on haematological parameters, such as haemoglobin (Hb), red blood cells (RBCs), packed cell volume (PCV), mean corpuscular volume (MCV), red cell distribution width (RDW), mean corpuscular hemoglobin concentration MCHC, polymorphs and lymphocytes values [158]. In an animal breast cancer model, MH was found to have the same anticancer activities as TH [163]. The anticancer effect of TH was successfully demonstrated in another three further in vitro studies utilizing the human breast cancer cell line [164,165,166]. These studies have found that TH treatment triggered caspase-3, caspase-7, and caspase-9 and decreased mitochondrial membrane potential in all tested cancer cells [164,165]. Moreover, TH was found to be cytotoxic to breast cancer cell line (MCF-7); it also protected non-tumorigenic epithelial breast cell line (MCF-10A) from the toxic effects of tamoxifen active metabolite 4-hydroxytamoxifen by increasing the efficiency of the DNA repair mechanism in these cells, as evidenced by the increment in DNA repair proteins Ku70 and Ku80 [166]. In a clinical trial, the combination of TH and anastrozole was more successful than anastrozole alone in decreasing breast background parenchymal enhancement in breast cancer patients [167]. In an animal model, TH revealed chemopreventive properties by decreasing cancer cell proliferation and activity, as seen by a reduction in the expression of cancer-related genes such as CCND1, EGFR, and COX-2. Furthermore, TH also reduced the expression of TWIST1 and RAC1, which are the genes representing epithelial-to-mesenchymal transition (EMT), and overexpressing β-catenin and E-cadherin [168]. Another study found that TH inhibits the growth of oral squamous cell carcinoma and osteosarcoma cell lines by causing early apoptosis [169]. The in vitro studies have revealed further anticancer potentials of TH, including anticancer activity against cervical cancer cell lines; this activity has the same mechanism as the in vitro research on breast cancer cell lines detailed above [164]. Previous research found that TH has antileukemic properties due to its potential to induce apoptosis in acute and chronic myeloid leukemia cell lines [170]. Another study showed that TH protected keratinocytes against UV radiation-induced inflammation and DNA damage by modulating the early biomarkers of photocarcinogenesis, thus providing significant protection from the adverse effects of ultraviolet B (UVB) radiation and a suggestion for its photochemopreventive potential [171]. Ramasamy et al. (2019) showed that after four and eight weeks of treatment with TH or Vitamin C, the fatigue level for experimental group was better than in the control group and no significant changes were detected between the vitamin C and TH groups for the white cell count and C-reactive protein [172]. Another study revealed that TH has anticancer capabilities; increasing the concentration of the extract reduces the viability of cancer cells [173]. Anticancer properties were also exhibited by KH. A previous study demonstrated that KH possesses chemopreventive properties in colorectal cancer-induced rats and also was found not toxic towards the animals [93]. Previous research indicated that KH was not cytotoxic to Human Gingival Fibroblast Cell Line (HGF-1 cell line) [174]. Salim et al. (2019) demonstrated how KH acts as a potential anticancer agent against human lung adenocarcinoma epithelial cell line (A549) and KH was capable of inducing cell cycle arrest at the G2/M phase [175]. A previous study showed that SH has anticancer activity and inhibited HepG2 cancer cell line proliferation [150]. Almeer et al. (2018) reported that the anticancer effects of SH correspond to their ability to modulate gene expression of MMPs and TIMPs in human breast adenocarcinoma (MDA-MB-231) cell lines [176]. SH was shown to have potential antitumor activity in studies on cancer cell growth inhibition [177]. Another study has shown that treatment with SH was able to inhibit proliferation, and induce apoptosis in breast cancer cell lines (MDA-MB-231 and MCF-7) and cervical cancer cell lines (Hela) [178]. Honey has a variety of phytochemicals with significant phenolic and flavonoid content, which contributes to its antioxidant properties [179]. Because free radicals cause oxidative stress, which leads to cancer formation, a drug with significant antioxidant capabilities might potentially prevent cancer [180]. Furthermore, a different type of polyphenol present in honey has been shown to have anticancer capabilities against a variety of cancers via various mechanisms. These polyphenols including, caffeic acid, caffeic acid phenyl esters, chrysin, galangin, quercetin, kaempferol, acacetin, pinocembrin, pinobanksin and apigenin [181]. Honey is a natural immune booster, antibacterial agent, anti-inflammatory agent, promoter of chronic ulcers and wound healing, and antioxidant; all of these characteristics contribute to its anticancer properties.

### 4.4. Antidiabetic Properties

There is evidence that honey has a beneficial effect on type 1 and type 2 diabetes mellitus. The measurement of fructosamine, glycosylated hemoglobin, and glucose is prevalent and important in the practice of glycemic control in patients with diabetes mellitus [183]. Previous studies showed that MH promoted the complete healing of diabetic foot ulcers, and decreasing the rate of minor amputations [184]. Studies using TH demonstrated a mild hypoglycemic impact, an enhanced liver enzyme profile, and a synergistic benefit on glycemic and metabolic profiles when taken in combination with metformin or glibenclamide [81,185,186]. Robert et al. (2009) reported that TH was found to have intermediary glycemic index values of 65 ± 7 [187]. Another study by Hussain et al. (2012) demonstrated that supplementation of twenty gram per day of TH for 4 months in healthy postmenopausal women caused a significant increase in fasting blood sugar (FBS)[188]. However, extending the therapy to 12 months resulted in a reduction in FBS levels in both healthy and diabetic postmenopausal women [189]. TH showed that the presence of flavonoid and phenolic compounds in honey from different botanical origins have strong α-amylase and α-glucosidase inhibition abilities as their inhibition percentages were more than 70.00% at 100 µg/mL and also recommended the uses of stingless bee honey in diabetes treatment [29,43,52,183]. Previous research revealed that honey had a varied effect on glucose metabolism for short- and long-term consumption [189]. Rashid et al. (2019) found that consuming 30 g KH for 30 days had no significant influence on fasting glucose and fasting lipid profile in patients with impaired fasting blood glucose [59]. The authors suggested that the small effect was due to short-term use of KH. In contrast, one in vivo study used forty male Wistar rats (divided into 5 groups; 8 weeks (C8) and 16 weeks control groups (C16), groups that received High-Carbohydrate High Fructose (HCHF) diet for 8 weeks (MS8) and 16 weeks (MS16), and a group that received HCHF for 16 weeks with KH supplemented for the last 35 days (KH) found that KH has the ability to normalise blood glucose and serum fasting blood glucose decreased in the KH group compared to the MS16 group [190]. Alharbi et al. (2022) reported that a combination of *L. plantarum*-fermented oats or fermented oats with SH exhibited potential antidiabetic effects and served as a potential approach for controlling glucose levels and lipid profiles, as well as protecting against oxidative stress [191]. Another study revealed that MH did not affect the weight in diabetic mice, whereas the SH-treated mice showed a reduction (7.7 ± 0.41 g) in body weight (Initial body weight: 18 ± 2.107, After treatment: 10.3 ± 2.517 and change: ↓7.7 ± 0.410) [192]. Clinical research has indicated that, unlike other sweeteners, honey consumption lowers the postprandial glycemic response in diabetic and non-diabetic volunteers, reducing the glucose blood concentration in patients with type 1 and type 2 diabetes [193]. Numerous studies attribute honey’s antidiabetic and hypoglycemic properties to its antioxidant capacity in response to dosage; indeed, the etiology of diabetes mellitus, particularly type 2 appears to be intimately linked to the presence of oxidative stress and ROS in numerous organs and tissues [194]. Increased glucose absorption by adipose tissue and muscles increases ROS generation, which contributes to oxidative stress, a process that regulates glycogen synthesis and glucose uptake. Furthermore, oxidative stress can promote insulin resistance by impairing the insulin signaling system, which can be reversed by honey therapy [195] Even in pancreatic -cells, oxidative stress has a role in affecting their functioning, resulting in inappropriate insulin secretion and an increase in -cell death. In general, it has been demonstrated that honey’s scavenger function reduces pancreatic oxidative stress [196]. Diabetes mellitus also compromises lipid metabolism, resulting in a significant presence of low-density lipoproteins (LDLs), which are oxidized and glycated in oxLDLs, causing endothelial damage. Even in this situation, honey’s antioxidant activity aids in the prevention of lipid oxidative metabolism in individuals with type 2 diabetes [96]. However, the antidiabetic efficacy of TH, KH, and SH remained unclear. Future research should extend understanding on the antidiabetic capabilities of TH, KH, and SH on diabetic patients to support the antidiabetic ability of TH, KH, and SH, as described in the animal studies above.

### 4.5. Antiobesity Properties

Honey provides various nutritional benefits and also honey has demonstrated the antiobesity effects in some studies [197]. Only KH has been shown in animal experiments to have antiobesity effects [198,199,200]. Previous study showed that supplementation of KH yielded a higher reduction in body mass index, the percentage of body weight gain, adiposity index, and relative organ weight in a high-fat diet-induced obese rat model than that of orlistat, an antiobesity drug [198]. Another study found that KH might prevent metabolic-syndrome-induced changes in omental fat mass, serum triglycerides, systolic blood pressure, diastolic blood pressure, adipocyte area, and adipocyte perimeter in rats [200]. However, honey showed better effects than orlistat, as the orlistat group showed hepatotoxicity effects. As of now, MH and TH have not been demonstrated to have an antiobesity effect. MH and TH have not yet been shown to have an antiobesity impact. Meanwhile, KH has not been studied for its antiobesity potential in humans.

### 4.6. Wound-Healing Properties

The importance of honey in the field of wound treatments has been well known since ancient times [95,139,201]. This practice was rooted primarily in tradition and folklore when investigators began to explore its medicinal potential [29,202]. There have been reports of significant therapeutic benefits of honey from various geographic regions on chronic wounds, ulcers, and burns [203,204]. Honey is reported to have excellent effects compared with traditional treatments for acute wounds, superficial partial-thickness burns, and infected wounds after surgery [205,206]. This healing ability is related to honey’s antioxidant and antibacterial properties, which help to keep wounds moist, as well as to honey’s high viscosity, which forms a barrier of protection over the wound and deters microbial infection. Its immunological activity has implications for healing wounds as well, acting both pro- and anti-inflammatory [103,139,207]. Normal wound healing is a difficult process made up of a number of overlapping activities (coagulation, inflammation, cell proliferation, tissue remodeling), in which the damaged tissue is eventually eliminated and replaced by reparative tissues [208]. While typical inflammation decreases when the neutrophil count falls within 1–2 days, the aggregation of these cells at the site of the wound causes an unbalanced network of regulatory cytokines, which results in a prolonged state of inflammation [209]. The majority of the bacteria in these chronic wounds exist as biofilms, which are a matrix of polysaccharides and other substances that prevent the use of antibiotics for wound healing [210,211,212]. Additionally, the challenge of treating chronic wound biofilm has worsened due to the increase in bacterial resistance to several antibiotics [213,214]. MH, TH, KH and SH have demonstrated to have wound-healing properties, as listed in Table 6. A study reported that MH eradicated methicillin-resistant *S. aureus* (MRSA) from colonized wounds and to inhibit MRSA in vitro by interrupting cell division. Additionally, MH restores MRSA susceptibility to oxacillin; molecular investigation revealed that it also affects the control of the *mecR1* gene, presumably explaining the recovered susceptibility [215]. Another study showed that a synergistic effect between rifampicin and MH was demonstrated on clinical *S. aureus* isolates, including MRSA strains. Unlike with rifampicin alone [216]. Therefore, it appears that MH has a potential to provide new synergistic combinations with antibiotics for treating wound infections caused by multidrug-resistant (MDR) bacteria. It is important to note that the antibiotics that have demonstrated a synergistic with MH belong to various antibiotic classes that inhibit various targets, including the 30S ribosome, RNA polymerase, membranes, and penicillin binding proteins. This supports the idea that honey is a complex substance perhaps with multiple active components that affect more than one cellular target site [215]. MH is also known to have a relatively low pH (3.5–4.5), which, in addition to preventing microbial development, induces macrophages to engage in bactericidal activity, and, in chronic wounds, decreases protease activity while increasing fibroblast activity and oxygenation [139,203]. Previous studies found that the topical application of TH on burn wounds contaminated with *P. aeruginosa* and *A. baumannii* provided a faster healing rate than chitosan gel or hydrofibre silver treatment [217]. Another study found that TH treatment improved wound contraction and *P. aeruginosa* growth control better than hydrofibre and hydrofibre silver treatment [218]. In the meantime, a different study found that administration of TH via oral administration improved anastomotic wound healing in rats with large bowel anastomosis by raising the proportion of fibroblasts and lowering the proportion of inflammatory cells, enhancing wound strength [219]. In a clinical trial looking into TH effect on promoting the post-tonsillectomy healing process has been conducted by comparing sultamicillin-treated group and sultamicillin + TH-treated group. During surgery, TH was applied topically, and for seven days later, 4 mL of it was taken orally three times each day. According to the results obtained showed that TH + sultamicillin-treated group’s healing process was much faster [220]. Based on the several research stated above, TH promotes wound healing more effectively than the standard antibiotic or topical treatment, which may support its usage as a wound dressing. Keloid scar formation is related to the healing of wounds. The excessive stimulation of EMT by TGFβ in keratinocytes results in the production of keloid scars. A study found that treating with KH reduced the TGFβ induced EMT in human primary keratinocytes, demonstrating the therapeutic potential of KH in decreasing the development of keloid scars [221]. Pretreatment with KH significantly reduced (*p* < 0.05) both the total area of ulcer and the ulcer index compared with the positive control group, and the percentage of ulcer inhibition in the KH pre-treated group was 65.56% compared with the positive control group. The treatment, KH, exhibited antiulcer properties against ethanol-induced gastric ulcer [222]. The study by Hananeh et al. (2015) showed the beneficial effect of SH on second-intention healing of full-thickness contaminated skin wounds in dogs. Another study reported that wounds treated with SH healed as fast as those wounds treated with iodine [223]. Another study demonstrated the activity of SH to induce burns in rabbits, and the areas of the wounds caused by the burns reduced from 3.38 mm^2^ to 2.36 mm^2^ after exposure to SH [224]. Honey is generally reported to have remarkable effects compared with conventional treatments for acute wounds, superficial partial-thickness burns and infected post-operative wounds [205,206].

### 4.7. Effects on Nervous System

Polyphenols are crucial to honey’s ability to protect the nervous system. Various brain diseases associated with aging can be countered by the scavenger activity against ROS, which are neurotoxic. Additionally, several age-related brain diseases are caused by the buildup of misfolded proteins such beta amyloid. This abnormal buildup can be prevented by polyphenols [225]. Previous studies demonstrated that MH alleviated Aβ-induced paralysis in the Alzheimer’s disease (AD) model CL4176 through HSP-16.2 and SKN-1/Nrf2 pathways [226]. Numerous research has shown that TH enhances memory and reduces depressive-like behavior in both humans and animals. In aged rats with noise-stress-induced memory problems, TH administration improved medial prefrontal cortex and hippocampus morphology, perhaps via lowering brain oxidative stress and increasing brain-derived neurotrophic factor (BDNF) concentration and cholinergic system activity [227]. Azman et al. (2015) reported that TH administration enhanced memory function and reduced depressive-like behavior in rats exposed to loud noise stress [228]. Another study on stressed ovariectomised rats revealed that TH or 17β-estradiol treatment has been shown to demonstrate antidepressive-like effects, possibly via restoration of the hypothalamic–pituitary–adrenal axis and enhancement of the BDNF concentration [229]. Al-Rahbi et al. (2014) showed that TH has an anti-anxiety effect in ovariectomized rats by improving their oxidative stress status [230]. In another study, rats administered with TH or 17β estradiol reported improved short-term and long-term memory and enhanced neuronal proliferation of hippocampal CA2, CA3 and dentate gyrus (DG) regions compared with untreated stressed ovariectomised rats [231]. According to recent study showed that TH pretreatment significantly attenuated an increase in lipid peroxidation level and decreased the total antioxidant status level induced by KA treatment in the rat cerebral cortex [232]. Previous studies showed that TH changed and altered in oxidative stress markers at the level of the spinal cord and neuronal damage in the spinal cord morphology of the offspring of prenatally stressed rats [233]. A study on metabolic-disease-induced rats showed that the KH-treated group exhibited less anxious behavior and demonstrated significant memory retention [190]. However, the underlying mechanism through which KH exhibits this effect has still not discovered to researchers. Within the literature, KH and SH have not been explored for their activity on the nervous system in humans.

### 4.8. Effects on the Cardiovascular System

Within the literature, only TH was reported to affect the cardiovascular system in three studies [234,235,236]: one combined in vivo/in vitro study [237] and two human studies [188,189]. TH was found to have a beneficial effect on the cardiovascular system in all animal studies; however, human studies showed varying degrees of results. Study by Khalil et al. (2015) found that the pretreatment of ischaemic rats with TH yielded significant protective effects on cardiac troponin, triglycerides and total cholesterol [234]. Two studies revealed that TH supplementation significantly lowered high systolic blood pressure by treating spontaneously hypertensive rats. [235,236]. Previous studies reported that TH could inhibit H_2_O_2_-induced vascular hyperpermeability in vitro and in vivo by suppressing adherence junction protein redistribution via calcium and cAMP [237]. A human study evaluated the effects of 20g of TH alone and a honey cocktail (a combination of TH, beebread, and royal jelly) for 12 months. When compared to supplementing with honey cocktail, TH showed a significant impact in lowering diastolic blood pressure and FBS and honey cocktail showed remarkable effects on body mass index. In addition, the results revealed that there are no significant changes in the blood pressure of healthy adults who were on royal jelly supplementation. However, no demonstrable effect of TH on the lipid profiles and anthropometric measurements was found and authors mentioned that further studies are required to formulate the best proportion of each bioactive compound in honey cocktail in order to exclude a possible matrix effect or antagonist effect of honey cocktail and to assess the physicochemical properties of TH and honey cocktail [189]. A randomised controlled trial comparing the effects of TH and hormone replacement therapy for 4 months revealed no demonstrable impacts on blood pressure measurement, BMI, and waist circumference. No significant difference was found in the lipid profile, blood sugar profile and bone density between the two groups [188]. Honey reduced fibrinogen levels and prevented coagulation via the intrinsic, extrinsic, and common coagulation cascades. For all of the reasons mentioned, honey can be considered as being very effective at preventing the development of atherosclerotic plaques, which can result in the onset of cardiac disorders. Lipid peroxidation is a major factor in the pathophysiology of atherosclerotic plaques as well [238]. It is confirmed that the honey phenolic compounds have a preventive and protective effect towards the damaging action of free radicals, counteracting, as explained, lipid peroxidation [239,240].

### 4.9. Effects on Respiratory System

The majority of scientific research on the protective effects of honey on the respiratory system is focused on asthma. A chronic inflammatory condition called asthma is defined by a generally reversible obstruction of the lower airways, which frequently results from allergen activity [241]. Previous studies reported that honey inhalation is able to reduce inflammation of the lower airways in a rabbit model of ovalbumin-induced chronic asthma [242]. Another study showed that the ingestion of honey in high doses (1 g/kg body weight daily for four weeks) improved the overall symptoms for up to one month after the end of treatment [243].

## 5. Conclusions

It has been extensively shown that MH, TH, KH, and SH provide a variety of health benefits for many different diseases and systems. Clinical studies, particularly those on wound healing, demonstrated that MH and TH are superior to standard dressings as a wound dressing. Clinical trials using TH, KH, and SH are still required, however, to add to the findings because convincing effects have not yet been finalised. The mechanism of action of these types of honey was also unclear, particularly in human studies, providing the need for additional research. In addition, TH, KH, and SH have been well researched compared with MH. Future research may, therefore, concentrate more on the potential benefits of KH and SH. This study also showed that, in comparison to MH, TH, KH, and SH have excellent preclinical potential in a variety of diseases and physiological systems, and this could guide future research into thoroughly examining honey as an effective and proven superfood that can be optimized for the benefit of humanity.

## Figures and Tables

**Figure 1 antibiotics-12-00337-f001:**
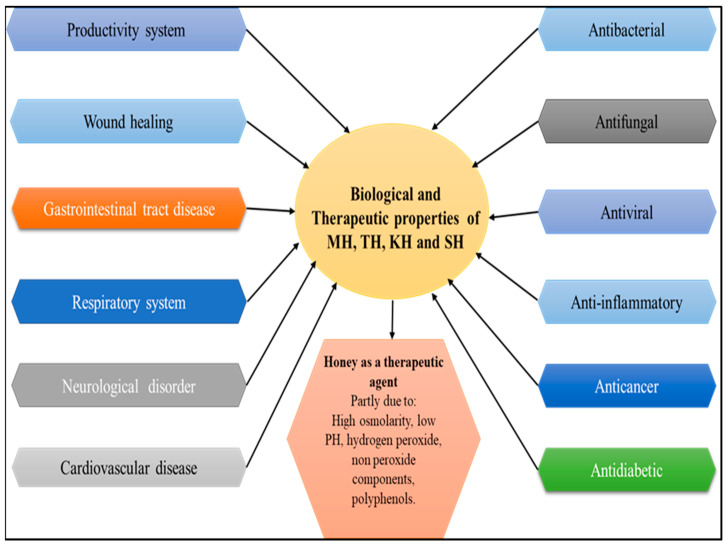
Schematic representation of the therapeutic effects of honey. Adapted from Nweze et al. (2019) [26], Vazhacharickal et al. (2021) [27], Al-kafaween et al. (2022) [28], and Rao et al. (2016) [61].

**Figure 2 antibiotics-12-00337-f002:**
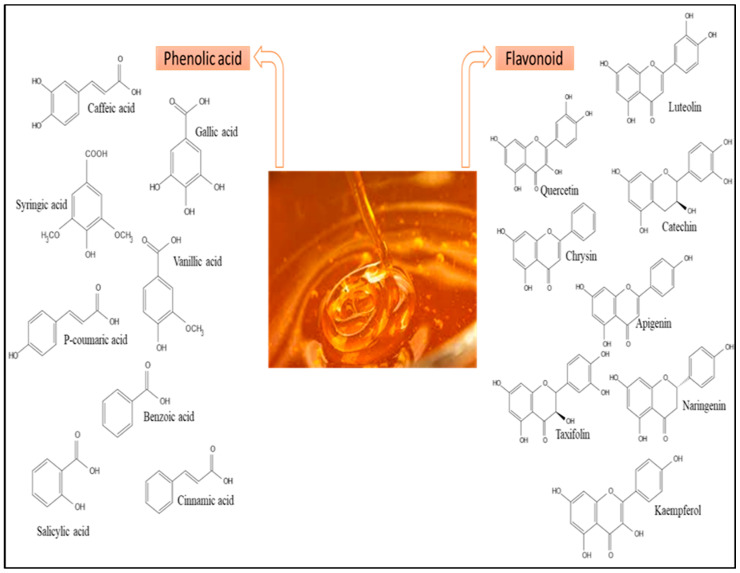
Most common phenolic compounds identified in honey. Adapted from Cianciosi et al. (2018) [67].

**Figure 3 antibiotics-12-00337-f003:**
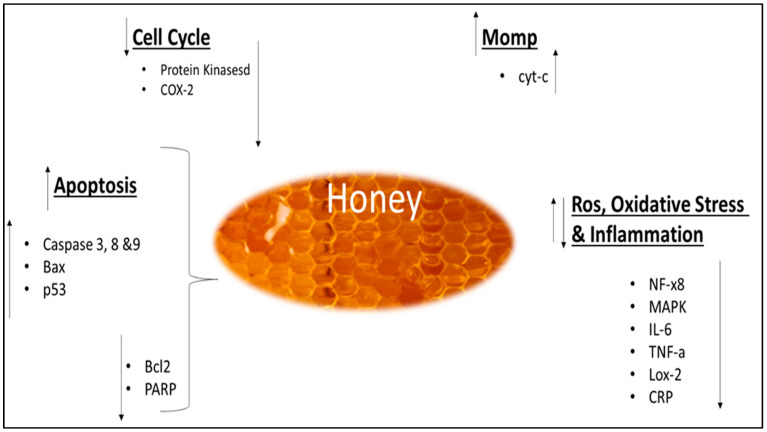
Molecular mechanism involved in anticancer effect of honey. Adapted from Cianciosi et al. (2018) [18] and Mohd et al. (2021)[29].

**Table 1 antibiotics-12-00337-t001:** The physicochemical characteristics of TH, KH, SH versus MH.

Physicochemical Properties	MH	TH	KH	SH
**Color appearance**	Light brown [43]	Dark brown [43]	Amber brown [29,43]	Light and dark honey [49]
**Moisture content (%)**	18.7% [41,43,50]	17.53–26.51% [29,35,41,51]	21.52–33.7% [29,43,52,53,54]	13.5–20.67% [11,31,55,56]
**Viscosity**	5.39–5.47 [57]	0.54–0.98 [57]	0.11–0.47 [57]	69.00 [31]
**Free Acidity (meq/kg)**	15.9–27.36 [58]	47.9–61.9 [57]	87.0–136.8 [57,59]	49.90 [31]
**PH**	3.20–4.21 [41]	3.14–4 [29,35,41,43,51]	2.76–4.66 [57,59]	3.90–5.2 [11,31,55,56]
**Ash content (g/100 g)**	0.03 [43,50,60]	0.19 [43,61,62]	0.01–0.66 [59]	0.68–2.33 [11,55]
**Water activity**	0.57 [57]	0.64 [57]	0.73–0.79 [57]	0.54 [56]
**Electrical conductivity** **(mS/cm)**	0.53 [43,50,60]	0.75–1.37 [35,43,51,62]	0.26–8.77 [57,59]	0.53 [11]
**HMF (mg/kg)**	48–400 [43,50,60]	46.17 [43,62]	8.80–69 [43]	18.48–31.74 [63]
**Apparent reducing sugar content (%)**	75.8–76% [41,43]	67.50–67.60% [41,43,61]	54.90–87.00 [43]	4.03–4.39 [11]
**Sucrose (g/100 g)**	1.66–2.98 [64]	0.01–1.66 [29]	0.01–1.26 [29]	0.7122–5.50 [55,65]
**Glucose (g/100 g)**	36.20 [61]	30.07–47–134 [29]	8.10–31.00 [29]	25.7–30.0 [55,65]
**Fructose (g/100 g)**	40.00 [61]	41.732–44.56 [29]	15–40.20 [29]	35.4–38 [55,65]
**Maltose (g/100 g)**	1.20 [61]	4.491 [29]	5.73–27.41 [29]	2.99–6.50 [65]
**Protein Content (g/kg)**	5.02–5.06 [64]	3.6–6.6 [29]	3.9–8.5 [29]	1.5–4.09 [55,65]
**Sodium (mg/kg)**	49.21–49.29 [66]	268.23–704.5 [29]	0.012–589.7 [29]	259.3 [55]
**Potassium (mg/kg)**	49.84–49.86 [66]	976.9–1576.40 [29,61]	0.07–732.2 [29]	2176.4 [55]
**Calcium (mg/kg)**	19.67–19.85 [66]	76.4–165.10 [29,61]	0.017–191.9 [29]	353.1 [55]
**Iron (mg/kg)**	49.04–49.14 [66]	11.17–128.13 [29]	6.5 [29]	114.7 [55]
**Magnesium (mg/kg)**	49.9–49.92 [66]	35.03–71.04 [29,61]	0.004–33.8 [29]	418.2 [55]
**Zinc (mg/kg)**	19.78–19.84 [66]	2.28–13.20 [29]	2.162 [29]	5.2 [55]
**Arsenic (mg/kg)**	9.38–9.44 [66]	0.015–0.062 [29]	0.019 [29]	Not detected [55]
**Lead (mg/kg)**	9.11–9.23 [66]	0.183–0.231 [29]	0.154 [29]	Not detected [55]
**Cadmium (mg/kg)**	9.87–9.91 [66]	0.004 [29]	0.002 [29]	Not detected [55]
**Copper (mg/kg)**	49.41–49.55 [66]	1.25–2.144 [29]	1.776 [29]	Not detected [55]
**Cobalt**	19.62–19.8 [66]	0.033–0.002 [29]	Not report found	Not detected [55]
**Total phenolic content (mg/kg)**	429.61 [64]	251.7–1103.94 [29]	477.30–614.7 [29]	212.4–520.34 [65]
**Total flavonoid content**	97.62 [64]	49.04–185 [29]	36.3 [29]	42.5 [67]

**Table 2 antibiotics-12-00337-t002:** The most common phenolic compounds (flavonoids and phenolic acids) identified in MH, TH, KH, and SH [18,41,68,69,70].

Phenolic Compounds	
Flavonoids	Phenolic Acid
**Manuka Honey**	
Chrysin (C_15_H_10_O_4_)	Caffeic acid (C_9_H_8_O_4_)
Galangin (C_15_H_10_O_5_)	Ferulic acid (C_10_H_10_O_4_)
Isorhamnetin (C_16_H_12_O_7_)	Gallic acid (C_7_H_6_O_5_)
Kaempferol (C_15_H_10_O_6_)	Syringic acid (C_9_H_10_O_5_)
Luteolin (C_15_H_10_O_6_)	-
Pinobanksin (C_15_H_12_O_5_)	-
Pinocembrin (C_15_H_12_O_4_)	-
Quercetin (C_15_H_10_O_7_)	-
**Tualang Honey**	
Apigenin (C_15_H_10_O_5_)	Caffeic acid (C_9_H_8_O_4_)
Kaempferol (C_15_H_10_O_6_)	Gallic acid (C_7_H_6_O_5_)
Luteolin (C_15_H_10_O_6_)	Syringic acid (C_9_H_10_O_5_)
Naringenin (C_15_H_12_O_5_)	Vanillic acid (C_8_H_8_O_4_)
Naringin (C_27_H_32_O_14_)	P-coumaric acid (C_9_H_8_O_3_)
Catechin (C_15_H_14_O_6_)	Benzoic acid (C_7_H_6_O_2_)
-	Trans-cinnamic acid (C_9_H_8_O_2_)
**Kelulut Honey**	
Luteolin (C_15_H_10_O_6_)	Gallic acid (C_7_H_6_O_5_)
Naringenin (C_15_H_12_O_5_)	Syringic acid (C_9_H_10_O_5_)
Taxifolin (C_15_H_12_O_7_)	Vanillic acid (C_8_H_8_O_4_)
-	3 4-dihydroxybenzoic acid (C_7_H_6_O_4_)
-	4-hydroxybenzoic acid (C_7_H_6_O_3_)
-	P-coumaric acid (C_9_H_8_O_3_)
-	Cinnamic acid (C_9_H_8_O_2_)
-	Salicylic acid (C_6_H_4_(OH)CO_2_H)
-	cis, trans-Abscisic acid (C_15_H_20_O_4_)
**Sidr Honey**	
Catechin (C_15_H_14_O_6_)	Gallic acid (C_7_H_6_O_5_)
Quercetin (C_15_H_10_O_7_)	Salicylic acid (C_6_H_4_(OH)CO_2_H)
-	Chlorogenic acid (C_16_H_18_O_9_)
-	Tannic acid (C_76_H_52_O_48_)

**Table 3 antibiotics-12-00337-t003:** Some of phenolic compounds with their potential health benefits found in MH, TH, KH, and SH.

Compound	Molecular Formulae	Potential Health Benefits	References
Apigenin	C_15_H_10_O_5_	Anti-inflammatory Antimutagenic Treating cardiovascular diseases	[18,41,43,68,69,70]
Caffeic acid	C_9_H_8_O4	Cardiovascular diseases treatment Anti-inflammatory effects Anticancer Antidiabetic	[18,41,43,68,69,70]
Catechin	C_15_H1_4_O_6_	Cardiovascular diseases treatment Antidiabetic potential Anti-inflammatory	[18,41,43,68,69,70]
Chrysin	C_15_H_10_O_4_	Improves cognitive deficits and brain damage Anticancer	[18,41,43,68,69,70]
Cinnamic acid	C_9_H_8_O_2_	Improves cognitive deficits and brain damage effect Antimicrobial effect	[18,41,43,68,69,70]
Gallic acid	C_7_H_6_O_5_	Antioxidant Anti-inflammatory Cardioprotective activity Antimutagenic Anticancer	[18,41,43,68,69,70]
Kaempferol	C_15_H_10_O_6_	Cardiovascular diseases treatment	[18,41,43,68,69,70]
p-Coumaric acid	C_9_H_8_O_3_	Anticancer activity Improves cognitive deficits and brain damage effect	[18,41,43,68,69,70]
Quercetin-3-O-rutinoside (rutin)	C_27_H_30_O_16_	Antiallergic Anti-inflammatory Antiproliferative Antitumor Cardiovascular diseases treatment	[18,41,43,68,69,70]

**Table 4 antibiotics-12-00337-t004:** List of micro-organisms that have been found to be sensitive to MH, TH, KH, and SH.

Type of Honey	Gram-Positive (G+) and Gram-Negative (G−) Bacteria	Concentration of Honey (% (*w*/*w*) or %, (*v*/*v*))	References
MH	**Gram-positive (G+) bacteria**	-	[24,41,104,112,116,130,131,133,134,135,136,137,138,139,140]
	*Streptococcus pyogenes*	11.25% (*w*/*v*)	
	*Coagulase negative staphylococci*	11.25% (*w*/*v*)	
	Methicillin-resistant *Staphylococcus aureus* (MRSA)	8.75% (*w*/*v*)	
	*Streptococcus agalactiae*	15% (*w*/*v*)	
	*Staphylococcus aureus*	11.25% (*w*/*v*)	
	*Coagulase-negative Staphylococcus aureus (CONS)*	10% (*w*/*v*)	
	*Staphylococcus epidermidis*	8% (*w*/*v*)	
	*Hemolytic streptococci*	6% (*w*/*v*)	
	*Enterococcus faecalis*	8% (*v*/*v*)	
	*Streptococcus mutans*	10.25% (*w*/*v*)	
	*Streptococcus sobrinus*	12.5% (*w*/*v*)	
	*Actinomyces viscosus*	11.25% (*w*/*v*)	
	*Bacillus subtilis*	16% (*w*/*v*)	
	*Bacillus cereus*	10% (*w*/*v*)	
MH	**Gram-negative (G−) bacteria**	-	
	*Stenotrophomonas maltophilia*	8.75% (*w*/*v*)	
	*Acinetobacter baumannii*	12.5% (*w*/*v*)	
	*Salmonella enterica Serovar typhi*	15% (*w*/*v*)	
	*Pseudomonas aeruginosa*	12.5% (*w*/*v*)	
	*Proteus mirabilis*	17.5% (*w*/*v*)	
	*Shigella flexneri*	17.5% (*w*/*v*)	
	*Escherichia coli*	17.5% (*w*/*v*)	
	*Enterobacter cloacae*	20% (*w*/*v*)	
	*Shigella sonnei*	6.61% (*v*/*v*)	
	*Salmonella typhi*	6–8% (*v*/*v*)	
	*Klebsiella pneumonia*	15% (*w*/*v*)	
	*Burkholderia cepacia*	5.2% (*w*/*v*)	
	*Helicobacter pylori*	5% (*v*/*v*)	
	*Campylobacter* spp.	1% (*v*/*v*)	
	*Porphyromonas gingivalis*	2% (*w*/*v*)	
	*Serratia marcescans*	9.4% (*v*/*v*)	
TH	**Gram-positive (G+) bacteria**	-	[41,112,115,116,130,138,141,142,143]
	*Streptococcus pyogenes*	12.5% (*w*/*v*)	
	*Streptococcus pneumoniae*	10% (*w*/*v*)	
	*Coagulase negative Staphylococci*	12.5% (*w*/*v*)	
	Methicillin-resistant *Staphylococcus aureus* (MRSA)	12.5% (*w*/*v*)	
	*Streptococcus agalactiae*	20% (*w*/*v*)	
	*Staphylococcus aureus*	10% (*w*/*v*)	
	*Staphylococcus hominis*	15% (*w*/*v*)	
	*Streptococcus haemolyticus*	12.5% (*w*/*v*)	
	Coagulase-negative *Staphylococcus aureus* (CONS)	12.5% (*w*/*v*)	
	*Staphylococcus epidermidis*	22.5% (*w*/*v*)	
	*Enterococcus faecalis*	12.5% (*w*/*v*)	
	*Bacillus subtilis*	15% (*w*/*v*)	
	*Bacillus cereus*	15% (*w*/*v*)	
TH	**Gram-negative (G−) bacteria**	20% (*w*/*v*)	
	*Stenotrophomonas maltophilia*	8.75% (*w*/*v*)	
	*Acinetobacter baumannii*	11.25% (*w*/*v*)	
	*Salmonella enterica Serovar typhi*	15% (*w*/*v*)	
	*Pseudomonas aeruginosa*	12.5% (*w*/*v*)	
	*Proteus mirabilis*	20% (*w*/*v*)	
	*Proteus vulgaris*	15% (*w*/*v*)	
	*Shigella flexneri*	20% (*w*/*v*)	
	*Escherichia coli*	20% (*w*/*v*)	
	*Enterobacter cloacae*	25% (*w*/*v*)	
	*Enterobacter aerogenes*	30% (*w*/*v*)	
	*Shigella sonnei*	4.9% (*w*/*v*)	
	*Salmonella typhi*	20% (*w*/*v*)	
	*Klebsiella pneumonia*	10.5% (*w*/*v*)	
	*Salmonella typhimurium*	12.3% (*w*/*v*)	
	*Salmonella enterica*	12.5% (*w*/*v*)	
	*Pseudomonas keratitis*	20% (*w*/*v*)	
KH	**Gram-positive (G+) bacteria**	-	[96,99,112,130,144,145,146,147]
	*Streptococcus pyogenes*	20% (*w*/*v*)	
	*Staphylococcus hominis*	6.25% (*w*/*v*)	
	*Streptococcus haemolyticus*	5% (*w*/*v*)	
	*Streptococcus agalactiae*	10% (*w*/*v*)	
	*Staphylococcus aureus*	20% (*w*/*v*)	
	*Streptococcus pneumonia*	20% (*w*/*v*)	
	*Bacillus cereus*	20% (*w*/*v*)	
	Methicillin-resistant *Staphylococcus aureus* (MRSA)	25% (*w*/*v*)	
KH	**Gram-negative (G−) bacteria**	-	
	*Pseudomonas aeruginosa*	20% (*w*/*v*)	
	*Escherichia coli*	20% (*w*/*v*)	
	*Klebsiella pneumonia*	10% (*w*/*v*)	
	*Salmonella* sp.	7.5% (*w*/*v*)	
	*Salmonella typhi*, *Shigella sonnei*	7.5% (*w*/*v*)	
	*Acinetobacter baumannii*	5% (*w*/*v*)	
	*Enterobacter clocae*	7.5% (*w*/*v*)	
	*Enterobacter aerogenes*	7.5% (*w*/*v*)	
	*Enterobacter aerogenes*	12.5% (*w*/*v*)	
	*Proteus mirabilis*	7.5% (*w*/*v*)	
	*Proteus vulgaris*	5% (*w*/*v*)	
SH	**Gram-positive (G+) bacteria**	-	[107,140,148,149,150,151,152,153,154,155,156]
	*Streptococcus pyogenes*	20% (*w*/*v*)	
	*Staphylococcus aureus*	10% (*w*/*v*)	
	*Staphylococcus epidermidis*	12% (*w*/*v*)	
	*Bacillus subtilis*	10% (*w*/*v*)	
	*Streptococcus mutans*	25% (*v*/*v*)	
	*Mycobacterium phlei*	7.5% (*w*/*v*)	
	Methicillin-resistant *Staphylococcus aureus* (MRSA)	25% (*w*/*v*)	
	*Bacillus cereus*	40% (*v*/*v*)	
SH	**Gram-negative (G−) bacteria**	-	
	*Pseudomonas aeruginosa*	20% (*w*/*v*)	
	*Klebsiella pneumonia*	15% (*w*/*v*)	
	*Escherichia coli*	10% (*v*/*v*)	
	*Salmonella Typhi*	15% (*v*/*v*)	
	*Salmonella enterica* serovar Typhimurium	12.5% (*w*/*v*)	
	*Salmonella enteritidis*	10 (*w*/*v*)	
	*Neisseria meningitides*	30% (*v*/*v*)	
	*Shigella flexneri*	10% (*w*/*v*)	
	*Serratia marcescans*	30% (*v*/*v*)	
	*Proteus vulgaris*	20% (*v*/*v*)	

**Table 5 antibiotics-12-00337-t005:** List of anticancer properties of MH, TH, KH, and SH.

Type of Honey	Type of Study	Finding	References
MH	In vivo	After cells were incubated with different concentrations of MH (range 0.3–2.5%) for 24–72 h, MH ameliorated breast cancer (MCF-7) and murine melanoma (B16.F1), colorectal carcinoma (CT26), and other cancer cells that proliferate dose-dependently via mediated the activation of a caspase 9-dependent apoptotic pathway, leading to the induction of caspase 3, reduced Bcl-2 expression, DNA fragmentation and cell death. This inhibitory effect on cell viability was dependent on both MH concentration and total incubation time.	[160,182]
MH	In vitro	MH exhibited profound inhibitory effects on cellular growth by reducing the proliferation ability, inducing apoptosis and arresting the cell cycle in a dose-dependent manner. MH induced cell cycle arrest in the S phase in HCT-116 cells, and simultaneously, in LoVo cells, it occurred in the G2/M phase through the modulation of cell cycle regulator genes (cyclin D1, cyclin E, CDK2, CDK4, p21, p27 and Rb). The expression of p-Akt was suppressed, whereas the expression of p-p38MAPK, p-Erk1/2 and endoplasmic stress markers (ATF6 and XBP1) was increased for apoptosis induction.	[159]
MH	In vitro	MH ameliorated human breast cancer MCF-7 cells. MH showed dose-dependent cytotoxicity towards MCF-7 cells after 24-h treatment.	[161]
TH	In vivo	The treatment groups were kindly received daily doses of 0.2, 1.0 and 2.0 g/kg body weight of TH, TH ameliorated breast cancer by increasing the susceptibility of proapoptotic proteins; apoptotic protease activating factor-1 (Apaf-1) interferon-gamma (IFN-γ) interferon gamma receptor-1 (IFNGR1) tumor protein P53 (p53) and decreased the expression of anti-apoptotic proteins; tumour necrosis factor alpha (TNF-α), cyclooxygenase-2 (COX-2) and B-cell lymphoma-extra-large (Bcl-xL).	[29,158]
TH	In vivo	The treatment groups were kindly given 0.2, 1.0 or 2.0 g/kg body weight/day of TH, TH alleviated breast cancers in rats by reducing cancer cell growth and enhanced histological grading.	[29,162]
TH	In vivo	The treatment groups were kindly treated with TH 1000 mg/kg and 2000 mg/kg by oral gavage for 10 weeks, TH showed chemo-preventive activities in oral squamous cell carcinoma-induced rats by suppressing cancer cell proliferation and activity and preserving cellular adhesion.	[29,168]
TH	In vitro	TH promoted apoptotic cell death induced by tamoxifen in breast cancer cell lines.	[29,165]
TH	In vitro	TH showed an anti-proliferative effect on oral squamous cell carcinoma and osteosarcoma cell lines by inducing early apoptosis.	[29,169]
TH	In vitro	TH demonstrated cytotoxic and apoptotic activities against human breast and cervical cancer cell lines with the mitochondrial apoptotic pathway’s involvement.	[29,164]
TH	In vitro	TH demonstrated apoptosis-inducing ability for acute and chronic myeloid leukaemia (K562 and MV4-11) cell lines.	[29,170]
TH	In vitro	TH protected keratinocytes from ultraviolet radiation-induced inflammation and DNA damage via modulation in early biomarkers of photocarcinogenesis.	[29,171]
TH	In vitro	TH was found to be cytotoxic to breast cancer cell line (MCF-7) but protected non-tumorigenic epithelial breast cell line (MCF-10A) from the toxic effects of tamoxifen active metabolite 4-hydroxytamoxifen.	[29,166]
TH	Human study	The treatment groups were kindly received oral TH 20 mg daily for 8 weeks, TH improved cancer-related fatigue and quality of life of patients with head and neck cancer post-chemotherapy or radiotherapy	[29,172]
TH	Human study	Combination of TH honey 20 g daily and anastrozole 1 mg daily showed more improvement in decreasing breast background parenchymal enhancement in patients with breast cancer than anastrozole alone.	[29,167]
TH	In vitro	TH has anticancer capabilities; increasing the concentration of TH reduces the viability of cancer cells	[173]
KH	In vivo	The treatment groups were kindly given oral administration of KH (1183 mg/kg body weight) twice daily for 8 weeks, KH possessed chemo-preventive properties in rats induced with colorectal cancer and also was found not toxic towards the animals.	[29,93]
KH	In vitro	KH was not cytotoxic to Human Gingival Fibroblast Cell Line (HGF-1 cell line)	[174]
KH	In vitro	KH possessed anticancer potential against human lung adenocarcinoma epithelial cell line (A549) as it was capable of inhibiting the cells growth in a dose and time-dependent manner. Moreover, KH was capable of inducing cell cycle arrest at G2/M phase.	[175]
SH	In vivo	The treatment groups were kindly given 20% of SH, SH has anticancer activity against HepG2 but not Hela cells. SH can be used as antimicrobial agent, but can be used as anticancer agent with care as it stimulated cell growth of some lines (e.g., Hala) and inhibited another (e.g., HepG2).	[150]
SH	In vitro	SH has possessed anticancer activity against breast adenocarcinoma (MDA-MB-231) cell lines and their ability to modulate gene expression of MMPs and TIMPs in human breast adenocarcinoma (MDA-MB-231) cell lines	[176]

**Table 6 antibiotics-12-00337-t006:** Summary of wound-healing properties of MH, TH, KH, and SH.

Type of Honey	Type of Study	Findings	References
MH	In vitro	MH eradicated and inhibited methicillin-resistant *S. aureus* (MRSA) from colonized wounds by interrupting cell division.	[215]
MH	In vitro	Combination MH with rifampicin was stopped the appearance of rifampicin-resistant *S. aureus,* which was rapidly lost in the presence of rifampicin alone.	[216]
TH	In vivo	The treatment groups were received TH (0.2 mL), there was a 14% and 73% reduction in wound size by day 9 and day 15 in the TH-treated wounds; however, this increased by 106% and 24% by day 12 and day 15, respectively.	[217]
TH	In vivo	The treatment groups were received TH (0.1 mL/cm^2^) dressing on the first burn wound, hydrofibre on the second wound, and hydrofibre silver on the last wound.Wound size was found to be markedly reduced in the TH-treated wounds on day 3, 9 and 15. The wounds showed a reduction in size of 12.86% by day 3 from the original 100 mm^2^ in the TH-treated wounds. They further decreased in size of 33.94% by day 9 post-burn. The wound healing process was observed for up to 21 days. On day 21, TH-treated wounds in *P. aeruginosa* inoculated group and *A. baumannii* inoculated groups healed completely. The remaining wounds in *P. aeruginosa* inoculated group and *A. baumannii* inoculated groups and all the wounds in *K. pneumonia* inoculated wounds did not healed completely.	[218]
TH	In vivo	The treatment groups were given TH 1.0 g/kg every morning until day seven post operatively, oral treatment with TH enhanced anastomotic wound healing by increasing the number of fibroblasts and decreasing inflammatory cells towards increased wound strength.	[219]
TH	Human	The treatment groups were received 3 mL of TH intraoperatively followed by 4 mL of oral TH three times daily for seven days, TH has positive effect in enhancing healing process in post tonsillectomy patient. It is easy to use topically, safe to consume orally and available at low cost locally. Overall it can be used as an excellent adjunct therapy for post-operative patients.	[220]
KH	In vivo	The treatment groups were given KH (1183 mg/kg) twice daily for 30 consecutive days by oral administration and on day 31, the rats were induced with absolute ethanol (5 mL/kg) via oral administration after being fasted for 24 h and were sacrificed 15 min after the induction. Pretreatment with KH significantly reduced (*p* < 0.05) both the total area of ulcer and the ulcer index compared to the positive control group. The percentage of ulcer inhibition in the KH pre-treated group was 65.56% compared with the positive control group. The treatment, KH, exhibited antiulcer properties against ethanol induced gastric ulcer.	[222]
KH	In vitro	KH reduced TGFβ-induced EMT in human primary keratinocytes, indicating its therapeutic potential in preventing keloid scar formation.	[221]
SH	In vivo	The treatment groups were received 1 to 2 mL of SH twice a day for one week and then once daily until the end of the study period (28 d). SH showed a beneficial effect of SH on second-intention healing of full thickness contaminated wounds and also wounds treated by SH healed as fast as those wounds treated by iodine.	[223]
SH	In vivo	The treatment groups were treated with 500 mg of SH (0, 7, 14 and 21 days), SH was found to possess higher healing rate of wounds induced either by thermal or chemical methods, In general, SH could be employed as topical wound healing agent and was also proved that SH could be used as natural wound healing agents besides commercial synthetic analogs.	[224]

## Data Availability

Not applicable.

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
