# Peer review of "Physicochemical Characteristics and Bioactive Compounds of Different Types of Honey and Their Biological and Therapeutic Properties: A Comprehensive Review"

_antibiotics, 2023, doi:10.3390/antibiotics12020337_

Round 1

Reviewer 1 Report

Peer-review report of the review article (antibiotics-2142584)

The manuscript entitled, “Physicochemical Characteristics and Bioactive Compounds of Different Types of Honey and Their Biological Activity: A Comprehensive Review” is an excellent and comprehensive review submitted for publication in the journal “antibiotics.”

This manuscript needs a major revision before acceptance.

The identification of substances having meliorating effects on human health is a great need of the time and has a great potential to attract scientists and related groups. Keeping in view the importance of this topic, the review report of the aforementioned article is as follows.

The Idea: The idea of this review article is well-conceived and publishable. However, the facts and research are presented in a broad and generic way. The specificity of presenting the facts is a key factor that must be considered in compiling past research in one manuscript.

Abstract: The abstract is well-written, presenting a good summary of the whole manuscript; however, it lacks to indicate the most specific features of each type of honey. 

Introduction: This review is structured to fulfill the requirements of a review article. In the introduction, background specificity regarding the synopsis of the work is well presented.

The method of review is well-written; however, a few problems are present.

Section 2. It is mentioned, “Several online databases were queried, including Web of Science, Scopus, Science Direct, PubMed and Google Scholar.” Is google scholar a database or a search engine?

Section 2. It is mentioned, “inclusion criteria for articles to be considered for this comprehensive review: honey antibacterial, antibiofilm, antiviral, antifungal, antioxidant, anti-inflammatory, anticancer, antidiabetic, gastrointestinal, 

respiratory, and cardiovascular and nervous system.” Was the word "honey" used with all of the keywords or just "antibacterial?" if you search a keyword "anticancer" without adding honey, numerous articles will pop up, how did you exclude them. Please clarify this. 

Physicochemical properties and Composition of MH, TH, KH and SH

Section 3. “Apies mellifera,” name of the species not italicized. 

Section 3. “Honey includes elements like organic acids and minerals.” Which organic acids and minerals? Although these are given in the table below, it is better to mention the most significant ones in the text. Tables should be for reference, but they do not fulfill the purpose of elaborating facts. 

Section 3. “Honey's electrical conductivity should not be higher than 0.8 mS/cm.” As per the rule, EC must not be higher than 0.8 sM/cm. The given figure (0.26-8.77 mS/cm) is many folds higher. How will it be justified in terms of the quality and safety of the honey?

Section 3. “Honey of acceptable grade need to have a moisture level of no more than 20 g/100 g.” What is the highest moisture content in the selected kinds of honey?

Section 3. “Neither yeast nor bacteria can grow at the pH levels that honey has.” The upper limit of the honey is 5.2, which is well inside the growth range of molds and yeasts. Then why they cannot grow in honey? It seems other factors hinder the growth, not the pH alone.

Section 3. “SH is light and dark honey.” Is it both dark and light depending on the factors? Or the same bottle is light and dark at the same time?

Section 3. “A total of six phenolic acids (syringic, gallic, benzoic, trans-cinnamic, p-coumaric, and caffeic acids) and five flavonoids (catechin, naringenin, kaempferol, luteolin, and apigenin) are found in TH.” What about the other types of honey included in this review article?

Section 3. “Hydrocarbons constitute more than half (58.5%) of its composition.” What does “its” refer to in this sentence?

Oxidative stress, antioxidant and anti-inflammatory properties

Section 4.1. “Oxidative stress causes oxidative damage, which can impair a variety of physiological activities.” Name some of the key physiological activities.

Section 4.1. “MH and TH contains a significant number of phenolic compounds.” Which phenolic compounds?

Section 4.1. “The level of malondialdehyde and glutathione peroxidase activity in the liver.” Which one was better, or both of them were equal?

Section 4.1. “TH has a hypoglycemic effect and lowers high MDA levels.” What is the level or value of lowering?

Section 4.1. “May have a hypoglycemic effect by restoring SOD and CAT activity.” To what level the SOD and CAT activity was restored?

Section 4.1. “Previous study showed that MH has higher antioxidant potential.” How much higher potential does it have than the other types of honey?

Section 4.1. “KH improved the glucocorticoid-induced osteoporosis by lowering lipid peroxidation and raising SOD levels.” How much did it lower, and what was the possible mechanism?

Section 4.1. “SH is a good source of antioxidant and biochemical components.” Which components are present?

Section 4.1. “A study done by Asari et al., (2019) revealed that the concentrations of TNF-α, IL6, and IFN-γ…” What is the efficient level of concentration of the aforementioned compounds?

Section 4.1. “Study by Minden et al., (2020) reported that 0.5% MH significantly increased the release of…” Was the honey taken as a solution in water? Or with food?

Antibacterial, Antiviral and Antifungal properties

Section 4.2. “…which makes it importance in the medical and therapeutic field.” Should it be “which makes it important?”

Section 4.2. “MH was more effective than other honeys both Gram-positive and Gram-negative bacteria.” How much more effective was MH than the other types of honey?

Section 4.2. “Honey has been known to have antiviral effects since the 19th century.” Does it kill the virus or hinder the viruses’ growth?

Section 4.2. “A previous study showed that the anti-HIV-1 activity of methylglyoxal was significantly more…” What is methylglyoxal?

Section 4.2. “Effect of MH on the HIV-1 RT activity is mediated by multiple constituents with different physical and chemical properties.” Explain the involved physical and chemical properties, and mention multiple constituents.

Section 4.2. “…because they include a variety of non-peroxide ingredients…” Enlist or mention non-peroxide ingredients.

Anti-cancer properties

Section 4.3. “MH significantly increased proapoptotic activity…” What was the significance level?

Section 4.3. “…resulting in a reduction in the size of the tumor.” What was the reduction level?

Section 4.3. “Study by Ahmed et al., (2017) reported that TH treatment was  effect on  haematological parameters…” Should it be, “Study by Ahmed et al., (2017) reported that TH treatment was effective on haematological parameters…?”

Section 4.3. “TH was found to be cytotoxic to breast cancer cell line.” Does TH kill the cells or change them?

Section 4.3. “…TH protected keratinocytes against UV radiation-induced inflammation and DNA damage…” How does it protect keratinocytes?

Section 4.3. “…increasing the concentration of the extract reduces the viability of cancer cells.” Which or what extract are the authors referring to?

Section 4.3. “…a distinct type of polyphenol present in honey.” What is the name of the distinct type of polyphenol?

Anti-diabetic properties

Section 4.4. “…different botanical origins yielded different degree of α-amylase and α-glucosidase inhibition…” Which one was better in inhibition?

Section 4.4. “Rashid et al., (2019) found that consuming KH daily for 30 days…” What was the dose?

Section 4.4. “…one in vivo study found that consuming KH for the last 35 days…” What was the dose?

Section 4.4. “…the SH treated mice showed reduction (7.7 ± 0.41 g) in body weight…” What was the weight loss in percentage?

Wound-healing properties

Section 4.6. “…possibly including numerous bioactive components that have an impact on multiple cellular target sites.” Which bioactive components are present?

Section 4.6. “…the total area of the ulcer and the ulcer index were dramatically reduced…” How much reduction was there?

Section 4.6. “…the activity of SH to induce burns in rabbits after exposure to SH.” Did SH increase burning?

Effects on Nervous System

Section 4.7. “TH enhanced the oxidative stress status, spinal cord morphology and nociceptive behavior in…” Does TH enhance oxidative stress? Will it not make TH toxic?

Effects on Cardiovascular System

Section 4.8. “…the effects of 20 g of TH alone and a honey cocktail (a combination of TH, beebread, and royal jelly) for 12 months. When compared to supplementing with honey cocktail, TH showed a significant  impact in  lowering diastolic blood pressure and FBS.” Why are other types of honey not contributing to showing better results? Which special component of TH lowered diastolic blood pressure and FBS?

Facts and data inclusion

Throughout the manuscript, the quality of the selected kinds of honey is elaborated by words including, “significant, dramatic increase or decrease, efficient,” and others. However, the quality is seldom proven by data and figures. Several examples are mentioned above. Every property or mechanism should be explained or confirmed by a specific figure, name, or mechanism. Recheck the whole manuscript, and add required figures, names, or mechanisms to make the claims believable.

The repetition of facts is present. If a few studies show similar results, it is better to merge them into one sentence. The whole manuscript must be concise and well-presented.

Formatting

The manuscript must be thoroughly formatted as several inconsistencies are found throughout the manuscript. 

A similar terminology or name must be used according to the author's guidelines. Several inconsistencies are found regarding names. A consistent name for the same substance must be used. 

The names of the kinds of honey are inconsistently formatted. Somewhere these are bold (MH), and somewhere normal. 

The sentences are starting with abbreviations, which must be rewritten to start the sentence with a proper word. 

The terms are inconsistently used, such as anticancer and anti-cancer. Check all the terms and use one format throughout the manuscript. 

The kinds of bacteria are written as Gram-negative and Gram-positive. These should be changed to gram- negative and gram-positive.

Language: The language and grammar of the manuscript must be improved.

Author Response

Dear reviewer,
Thank you very much for your consideration, and we really appreciate the comments and have learned a lot. Appropriate changes were made in the revised manuscript according to the suggestions of reviewers.

Reviewer 2 Report

Comment:

1.     Figure 1 typo hight instead of high

2.     Missing words in the sentences such as anti-cancer activity of honey.

Reference section is poorly presented, needs same font, either upper case or lower case, there should be uniformity thought the references.  

Author Response

Dear reviewer,
Thank you very much for your consideration, and we really appreciate the comments and have learned a lot. Appropriate changes were made in the revised manuscript according to the suggestions of reviewers 

1. Figure 1 typo hight instead of high: it has been corrected
2. Missing words in the sentences such as anti-cancer activity of honey. It has been corrected throughout MS
3. Reference section is poorly presented, needs same font, either upper case or lower case, there should be uniformity thought the references. It has been corrected

Reviewer 3 Report

In general some parts of the manuscript are very well discussed and are appropiatte. However other parts are lacking a lot and must be improved for being considered for publication. Please see in the attached pdf some comments.

Best regards!

Author Response

Dear reviewer,
Thank you very much for your consideration, and we really appreciate the comments and have learned a lot. Appropriate changes according to your suggestions were made inside the MS. Please see the attached file.

Round 2

Reviewer 1 Report

Peer-review report of the review article (antibiotics-2142584)

The manuscript entitled, “Physicochemical Characteristics and Bioactive Compounds of Different Types of Honey and Their Biological Activity: A Comprehensive Review” is an excellent and comprehensive review submitted for publication in the journal “antibiotics.”

The authors have significantly improved the manuscript; however, the manuscript is not up to the mark for publication. This manuscript still needs major revision before acceptance.

Overall Evaluation: The manuscript is about various types of honey, which should present the facts accordingly. In most sections, generic details about honey are given without attributing to a specific type. If the authors aim to present an overall image of honey, the method of facts presentation should be adapted accordingly.

The coordination among the headings is missing. Some facts repeat in each section.

The author’s response to the reviewer’s comments from revision one is not properly addressed. The line number tagging is not proper. The given line numbers do not show the addressed comments.

A rigorous revision of the whole manuscript is required to fix the facts presentation and their internal coherence.

Method of review

Lines 95-96. “A literature search was performed by combining the following set of keywords:

MH, TH, KH, and SH. Did you use “MH, TH, KH, and SH” as keywords? It is surprising because managing specificity by just a few alphabets is very hard. Can the authors provide specific keywords that they used?

Physicochemical properties and Composition of MH, TH, KH and SH

Lines 109-123. The composition of honey in general is given. No specificity regarding MH, TH, KH, and SH is found. 

Line 125-130. The standards are given. However, the values in MH, TH, KH, and SH specifically are absent. 

Lines 134-135. What is the pH Range of MH, TH, KH, and SH? 

Line 136-137. “…The pH range for honey was reported to be 2.76 to 5.2 [29, 41, 55, 28]” The range of fungi and bacteria is 4.0-4.5, which is inside the pH range of honey. Why microorganisms cannot grow in honey? 

Lines 152-153. “…and sex flavonoids (apigenin, kaempferol, luteolin, naringenin, naringin and catechin) are found…” What does “sex flavonoid” mean?

Table 1. What is the unit of minerals given in the table? 

Oxidative stress, antioxidant and anti-inflammatory properties

Line 237-239. “…their level of total phenolic content (TFC), total flavonoid content (TFC), radical-scavenging activity, ascorbic acid as well as total carotenoid content (TCC)…” Please give the values.

Line 245-256. “…A study done by Asari………… under chronic stress conditions…” There is no specificity regarding the use of honey. Elaborate on the study in line with the present study. Antibacterial, Antiviral and Antifungal properties

Line 301-304. Does honey kills all gram-positive and gram-negative bacteria? 

Line 323-339. “…A previous study showed that monofloral honeys from……… chemical properties as mentioned above...” How are these facts linked with MH, TH, KH, and SH? Or this study aims to include every type of honey? 

Table 3. What is the sensitivity cutoff value? Are weakly sensitive species are included also? 

Language: The syntax, language and grammar of the manuscript must be improved.

Author Response

Dear reviewer,
Thank you very much for your consideration, and we really appreciate the comments and have learned a lot. Appropriate changes were made in the revised manuscript according to the suggestions of reviewers.

Note: We put another table about phenolic compounds in honey, (general), please if you agree, we can put it.

Reviewer 3 Report

Authors have appropiately corrected some of the reviewer inquiries but not all of them.

-As the tittle work is "Physicochemical Characteristics and Bioactive Compounds of 2 Different Types of Honey and Their Biological and Therapeu-3 tic Properties: A Comprehensive Review", the physicochemical characteristics have to be dicussed more. This part is still lacking the most.

Anyways find below other previous reviewer comments that should be at least addresed:

-Figure 2 does not shown any comparison. It is listing the physicochemical properties of honey but in my opinion it is not necesary. If authors would like to keep it please include at least ranges in those parameters. However this informtaion is given in Table 1.

-If the sugar composition varies according to the floral source please give a range after fructose and glucose and not an exact number as it is given

-Include the concentration at which each microorganism is reduced by each honey in table 3.

-Please, include in the table 4 and 5 at least the specific treatment (doses, times...) for the in vivo studies.

In my opinion this work could be accepted after major revisions.

Author Response

(The authors gave the same response as above.)

Round 3

Reviewer 1 Report

The authors have significantly improved the manuscript.
The manuscript is acceptable in its current form after proofreading for grammar and English language.

Reviewer 3 Report

Authors have addressed the reviewer comment appropriately. Best regards!